# The Stigma and Infertility-Related Stress of Chinese Infertile Women: A Cross-Sectional Study

**DOI:** 10.3390/healthcare12111053

**Published:** 2024-05-21

**Authors:** Dan Luo, Yi-Bei Zhouchen, Lu Li, Yu-Lei Jiang, Yi Liu, Sharon R. Redding, Rong Wang, Yan-Qiong Ouyang

**Affiliations:** 1School of Nursing, Wuhan University, Wuhan 430071, China; 2018283050122@whu.edu.cn (D.L.); zcyb1228@whu.edu.cn (Y.-B.Z.); 2018303051032@whu.edu.cn (Y.-L.J.); 2The First Affiliated Hospital, Zhejiang University School of Medicine, Hangzhou 310003, China; 2019283050067@whu.edu.cn; 3Zhongnan Hospital of Wuhan University, Wuhan 430071, China; 2017203050037@whu.edu.cn; 4Project HOPE, Washington, DC 20036, USA; shredding@cox.net; 5Renmin Hospital of Wuhan University, Wuhan 430060, China

**Keywords:** infertile women, stigma, stress, influencing factors

## Abstract

Objectives: This study was conducted to investigate the stigma status of infertile women in China and to determine the influencing factors. Methods: 366 infertile women from the gynecological and reproductive departments of two tertiary hospitals completed socio-demographic questionnaires, the Infertility Stigma Scale (ISS) and the Mandarin Fertility Problem Inventory (M-FPI). Results: The scores of stigma and infertility-related stress in infertile women were (52.51 ± 17.74) and (150.03 ± 17.51), respectively. Multiple regression analysis found that location of residence, regarding children as the most important thing in life, talking to others about infertility and infertility-related stress were the main influencing factors of stigma in infertile women, which explained 17.3% of the total variance. Conclusions: In the current study, the level of stigma in women with infertility was at the middle range. Location of residence, regarding children as the most important thing in life, whether to talk with others about infertility and infertility-related stress were the four main influencing factors of stigma.

## 1. Introduction

The World Health Organization (WHO) defined infertility as a disease of the reproductive system, involving failure to achieve a clinical pregnancy after 12 months or more of regular unprotected sexual intercourse, which has become a global public health problem [1]. The prevalence of infertility is an indicator of reproductive health and medical service level, as well as the economic and cultural status of a community [2,3]. It is estimated that about 60 to 80 million couples in the world have difficulty in getting pregnant [4]. The WHO predicted that infertility would become the third major disease after cancer and cardiovascular disease in the 21st century [1]. In the United States, approximately 12.7% of child-bearing age females seek treatment for infertility each year [5]. In contrast, the infertility rate among Chinese couples of child-bearing age was 25% and increased with age [6]. 

Infertility is a multifaceted problem with physical, psychological, financial and social consequences [7,8]. Even now, motherhood is still regarded as the core of women’s status and a key factor in determining their status in the family and society. Regardless of which one of the couple the infertility originates from, the female tends to be blamed in society. More specifically, the reasons for female infertility were more often considered to be morally related behaviors, such as multiple sexual partners, abortion and use of contraceptives, while the reasons for male infertility were more often considered to be physical limitations [2]. The stereotypes and prejudices about infertility of the social, reproductive pressure of women and families, the economic burden of treatment and ineffective coping styles may lead to infertile women’s stigma, broken marriages and psychological disorders, such as anxiety and depression [9,10,11].

Stigma is defined as having a negative self-perception as a patient, becoming alienated and isolated in society, being insulted and not understood and having negative self-perceptions about the behaviors of others [12]. Scholars defined the stigma of women with infertility as actual or imagined fear of being humiliated or ostracized due to the undesirable nature of infertility, and women direct this fear at themselves, showing self-devaluation behaviors [13]. Stigma is prevalent in women with infertility. Women with infertility express negative emotions, such as anxiety, depression, self-devaluation, self-isolation and a general sense of powerlessness [10]. They tended to internalize the negative views of others toward themselves and described themselves as having ‘broken wings’, as being ‘half a woman, half a man’ and even disabled [14]. Among infertile women, stigma was significantly associated with quality of life, depression level, treatment enthusiasm and compliance, with higher levels of stigma predicting poorer quality of life, higher levels of depression and lower treatment initiative and compliance [15,16].

Currently, studies on infertility have focused on pathogeny [17], psychological distress [18] and interventions [19], while fewer have focused on infertility stigma, especially in China. With the increasing incidence of infertility, the stigma of infertile women is in urgent need of attention and targeted improvement. It is important to explore the status quo of stigma of women with infertility in China and propose reasonable and effective coping strategies. For this reason, this study will focus on the current status of stigma in infertile women and analyze the influencing factors. This provides evidence for healthcare professionals to assist infertile women in choosing appropriate treatment so as to relieve their physical and mental stress.

## 2. Methods 

### 2.1. Design and Sampling

From November 2018 to September 2019, women in reproductive clinics and gynecology departments of two large hospitals in Wuhan were selected via convenient sampling and invited to participate in this study. The inclusion criteria were: (1) women having a diagnosis of infertility; (2) at least 18 years of age; (3) able to read and write Mandarin Chinese. Women with complications, including cancer, mental illness or severe chronic disease were excluded. This study was approved by the ethics committee of the first author’s university and tertiary-level hospitals (2020YF0084), and participants were provided with informed consent. 

### 2.2. Measures 

#### 2.2.1. Socio-Demographic Questionnaire 

This questionnaire was designed by the researchers after a review of the literature and group discussion and included demographic and disease-related questions. Data collected included age, educational background, location of residence, number of children and annual household income. Infertility-related questions included whether they talked to others about infertility and whether they regarded children as the most important thing in life. 

#### 2.2.2. Infertility Stigma Scale (ISS) 

Based on Bem’s self-perception theory and Ellis’s ‘ABC’ theory, Fu and coworkers developed the ISS to measure the stigma of infertile women in China [20]. The ISS is a 27-item scale with four sub-scales, including self-devaluation (seven items), social withdrawal (five items), public stigma (nine items) and family stigma (six items). Each item uses a five-point Likert scale (1 = completely disagree, 2 = disagree, 3 = not sure, 4 = agree, 5 = completely agree). The total scores of the scale range from 27 to 135. The higher the scores, the higher the stigma level of women with infertility. The Cronbach’s α of this scale was 0.94.

#### 2.2.3. Mandarin Fertility Problem Inventory (M-FPI) 

In 1999, Newton and colleagues developed the Fertility Problem Inventory (FPI) to measure the fertility stress of infertile women [21]. Peng and colleagues [22] brought the FPI to China and investigated 223 infertile couples in a tertiary hospital in Harbin using a version in Mandarin (M-FPI) [22]. The results showed that the Cronbach’s α was 0.81, with good discriminant validity and convergent validity. The M-FPI includes 46 items in five sub-scales: social concern (ten items), sexual concern (eight items), relationship concern (ten items), need for parenthood (ten items) and rejection of childless lifestyle (eight items). Respondents are asked to indicate their degree of agreement with each item using a six-point Likert scale ranging from 1, being rated as strongly disagree, to 6, being rated as strongly agree. Among these 46 items, 18 items are reverse scored. The Cronbach’s α of this scale was 0.86.

### 2.3. Data Collection

Participants were informed that the survey was confidential and anonymous. Researchers explained the purpose of this study, the content of the questionnaires and how they should be completed. After participants completed the questionnaires, researchers validated if the questionnaires were completed appropriately.

### 2.4. Data Analysis

Analysis was completed using SPSS 25.0. Categorical data were summarized via frequency and percentage. Mean values and standard deviation were calculated to describe the scores of stigma and infertility-related stress. The differences between the mean values of groups were compared with *t*-tests and analysis of variance. With socio-demographic characteristics as the independent variable and the scores of stigma as the dependent variable, multiple linear regression analysis was used to assess the association. Larger adjusted R^2^ represents better model fit. All hypothesis tests were completed with 0.05 as a test level, and *p* < 0.05 was considered statistically significant.

## 3. Results

### 3.1. Socio-Demographic Characteristics of Participants

In this study, 480 questionnaires were distributed and 366 were deemed as valid for a return rate of 76.25%. Among the 366 infertile women, the average age was (31.7 ± 3.7) years. A total of 266 participants (72.68%) had college degrees or higher. More than half of the participants (65.85%) regarded children as the most important thing in life (see Table 1).

The average score and average item score for the total ISS were (52.51 ± 17.74) and (1.94 ± 0.66), respectively. The average scores of the four sub-scales were public stigma (15.69 ± 6.37), self-devaluation (13.66 ± 5.34), social withdrawal (12.31 ± 4.72) and family stigma (10.85 ± 4.54), from the highest to the lowest. The total scores of infertility-related stress was (150.03 ± 17.51). The average scores of the five sub-scales were social concern (36.75 ± 5.62), need for parenthood (31.91 ± 5.54), relationship concern (29.42 ± 5.43), sexual concern (26.37 ± 4.55) and rejection of childless lifestyle (25.59 ± 4.84), from the highest to the lowest. Details are shown below (see Table 2).

### 3.2. Comparison of Stigma Scores of Infertile Women with Demographic Characteristics

The results of *t*-tests and analysis of variance showed that factors including educational background, location of residence, annual household income (in CNY), regarding children as the most important thing in life, talking to others about infertility and infertility-related stress had significant effects on the level of stigma in women with infertility (*p* < 0.05) (see Table 1).

### 3.3. Predictive Factors of Stigma in Women with Infertility

Multiple regression analysis was conducted with statistically significant variables as independent variables and stigma scores as dependent variables. The variable assignment was as follows: location of residence: rural area = 0, urban area = 1; regard children as the most important thing in life: yes = 0, no = 1; talk to others about infertility: yes = 0, no = 1. The results of the regression model indicated that location of residence, regarding children as the most important thing in life, talking to others about infertility and infertility-related stress were the four main influencing factors of stigma in female women with infertility (*p* < 0.05), which explained 17.3% of the total variance. Participants living in urban areas had significantly lower stigma than those living in rural areas (*t* = −2.097, *p* = 0.037). Participants who regarded children as the most important thing in life had significantly higher stigma than those who did not (*t* = −3.702, *p* < 0.001). Participants who talked to others about infertility had significantly lower stigma than those who never discussed this issue with friends/family (*t* = 2.149, *p* = 0.032) (see Table 3).

## 4. Discussion

### 4.1. Mid-Range Level of Stigma in Women with Infertility

The current study suggested that infertile women had a medium level of stigma (52.51 ± 17.74), which is slightly lower than the study of Yilmaz et al. (60.79 ± 2.03) [16]. The possible reason is that only two hospitals were selected in this research, which limited the area of residence of the women, and the proportion of women with a college degree or above was much higher than in the study by Yilmaz et al. [16].

The top two of average item scores of the dimensions were social withdrawal and self-depreciation, which is consistent with the result of Zhang et al. [23]. Participants reported feeling embarrassed in social gatherings, especially when others knew they were ‘infertile’ or when children were the subject of the conversation [14]. They preferred avoiding such gatherings. Infertility is treated in health service institutions and the cost is high and treatment may last many months, or even years [24]. Due to the decrease in interpersonal communication, participants had fewer topics in common with the people around them, which resulted in a sense of alienation and exclusion. Women may suffer from stigma, social isolation and ridicule within their communities [10]. Therefore, women with infertility preferred to keep their infertility a secret and reduce or even avoid social contact, resulting in social withdrawal. Infertile women have a strong desire to have children and even regard it as the most important thing in their lives. Because of infertility, women expressed feelings of anxiety, frustration, grief, lack of self-esteem and a general sense of powerlessness [25]. The failure to achieve self-worth leads to women’s self-devaluation. Moreover, women were to blame for fertility problems in traditional social culture [26]. There is a common belief in Chinese society that “Childlessness is always the woman’s fault” [27]. Infertile women tended to have low self-esteem, anxiety and other negative feelings, even a strong sense of guilt and self-devaluation, thereby losing confidence in their life and marriage. At the same time, they feel that friends or family cannot understand them and leave them behind. 

### 4.2. Reasons for Influence of Related Factors

#### 4.2.1. Infertility-Related Stress 

In this study, the mean score of infertility-related stress was (150.03 ± 17.51), which was similar to the results of Jiang et al. [13]. The overall stress of infertile women was high and was mainly caused by relationship and social concerns. It results in the loss of identity as a woman or potential parent and tension in interpersonal relationships due to the desire but inability to get pregnant/give birth for females [28]. Women receiving assisted reproductive therapy have to face treatment and economic pressure. Making parenthood as a core goal of life may increase the influence of infertility in a person’s life. A previous study has shown that statements about the importance of parenthood can aggravate the depressive symptoms of infertile patients through a mediating factor of experiential avoidance [29]. As an emotion regulation strategy, although experiential avoidance can provide short-term emotional relief, it can also lead to negative psychological outcomes when the response is rigid or inflexible. Patients with high infertility-related stress often fail to integrate into normal social circles, and the care from relatives and friends will make them feel embarrassed and distressed [2]. Therefore, they tend to adopt the avoidance type of responses, which leads to the deepening of social withdrawal, the loss of social support and the aggravation of stigma.

#### 4.2.2. Talking to Others about Infertility 

Findings in the current study indicated that participants who talked to others about infertility had significantly lower stigma than those who never discussed this issue with friends/family. Participants who never discussed infertility were more socially withdrawn. They avoided interactions with people who might be prejudiced and kept their reproductive status a secret because attending social events reminded participants how different they were [30]. This resulted in a loss of social support due to a weakening of intimate relationships with others. Social support is an effective external resource, which is closely related to the individual’s mental health. Good social support can cushion against an individual’s acute state of stress, then weaken the adverse effects [31]. Infertile women received less support from friends than fertile women and were less likely to feel enough social support compared to fertile women [32].

#### 4.2.3. Regarding Children as the Most Important Thing in Life 

In this research, participants who regarded children as the most important thing in life had higher stigma. If a woman fails to reproduce, she may feel shame, because she could not fulfill her duty to the family [27]. Some involuntarily childless women also perceived themselves as incomplete and failed women [8]. Many participants considered children as their meaning of life and self-worth. Infertility caused adverse feelings in women, such as being deemed a symbol of bad luck, failure as a woman and a strong sense of self-devaluation, resulting in a high level of stigma.

#### 4.2.4. Location of Residence 

In the current study, participants living in rural areas had higher stigma. A study in Ghana showed that infertile women living under a traditional cultural environment would pay more attention to the importance of fertility [3]. In traditional culture, motherhood provides women with a powerful socio-cultural status that includes privileges; childless women are powerless and cannot easily confront these societal beliefs [14]. In many developing countries, social stigmas are attached to infertility [3]. Psychologically, women lose their ‘womanhood’ and sense of gender identity when they cannot conceive. In some cultures, women can experience having no children as discrediting, feeling that others perceive getting pregnant as natural and inevitable and women with no children as strange, deficient and incomplete [8]. Family members held a discriminative attitude to infertile women because they could not give birth to a child for the family, which was regarded as a shame. Even women with infertility shared this line of reasoning, which added to their mental health burden. This line of reasoning contrasts with the experience of residing in large communities and cities, where the level of general education is often higher and where residents may keep an open mind. Participants living in urban areas indicated they felt less moral and spiritual pressure and a lower level of stigma regarding their infertility. Therefore, it is difficult to attribute the cause of infertility to only women. 

### 4.3. Implications and Suggestions

Infertility can increase women’s psychological distress and negative emotions, and reduce adherence to infertility treatment, which leads to an increase in the difficulty of treatment. In addition, females have more pressure or burden about infertility than males. Therefore, reducing the stigma of women with infertility could not only maintain the mental health of infertile women, but also promote the treatment of infertility.

Education on infertility should be widely publicized to change people’s prejudices and help correct traditional beliefs of criticizing infertile women. Social support of women with infertility should be promoted by helping people understand their role in the stigma of infertile women and the assistance and comfort they can provide. Infertile women should be encouraged to communicate with others, using a variety of methods, such as online forums, blogs and support groups as well as counseling services. For women with excessive stigma, professional psychological treatment can be provided. Personalized psychological interventions can be designed and implemented to help women maintain mental health and reduce psychological stress.

### 4.4. Limitations

This research was conducted in one geographic area using convenient sampling, which limited the representativeness of the sample. A multicenter and large sample study should be implemented in the future. Second, the study did not include stigma in men with infertility. This may lead to one-sidedness of the results and aggravate the misunderstanding of infertile women, which is not conducive to improving the level of stigma among infertile women. Moreover, the failure rate of assisted reproductive therapy and the relationship with the partner can also affect the mental state of infertile women. Future research should incorporate more relevant factors.

## 5. Conclusions

In the current study, the level of stigma in women with infertility was at the middle range. Location of residence, regarding children as the most important thing in life, whether to talk with others about infertility and infertility-related stress were the four main influencing factors of stigma. Although this research showed that the stigma level of women with infertility was not high, the negative impact of stigma on infertility should not be ignored. Therefore, healthcare personnel should not only pay attention to the physical health status of infertile women, but also assess the stigma and mental health of women with infertility and address ways to reduce this stigma as much as possible. 

## Figures and Tables

**Table 1 healthcare-12-01053-t001:** Socio-demographic characteristics of participants and mean differences on stigma (N = 366).

Item	N (%)	Stigma (Mean ± SD)	*t*/*F*	*p*
Age (in years)			0.947	0.389
20–25	17 (4.64)	83.82 ± 25.71		
26–30	155 (42.35)	94.95 ± 28.76		
≥31	194 (53.03)	94.45 ± 34.68		
Education background			−2.803	0.005 *
High school or less	100 (27.32)	101.73 ± 30.25		
College or higher	266 (72.68)	91.33 ± 32.13		
Location of residence			3.615	<0.001 **
Rural area	158 (43.17)	101.10 ± 31.78		
Urban area	208 (56.83)	89.09 ± 31.13		
Annual household income (in CNY)			2.596	0.010 *
≤50,000	81 (22.13)	102.23 ± 33.98		
>50,000	285 (77.87)	91.88 ± 30.99		
Regard children as the most important thing in life			5.187	<0.001 **
Yes	241 (65.85)	100.20 ± 32.08		
No	125 (34.15)	82.56 ± 28.31		
Have other children			−0.641	0.522
Yes	73 (19.95)	92.03 ± 36.94		
No	293 (80.05)	94.70 ± 30.59		
Talk with others about infertility			−1.982	0.048 *
Yes	274 (74.63)	92.25 ± 29.94		
No	92 (25.14)	99.86 ± 36.94		

Note: CNY: Chinese Yuan. * *p* < 0.05, ** *p* < 0.001.

**Table 2 healthcare-12-01053-t002:** Scores of stigma and infertility-related stress (N = 366).

Item	Scores (Mean ± SD)	Score of Each Item (Mean ± SD)
Stigma	52.51 ± 17.74	1.94 ± 0.66
Self-devaluation	13.66 ± 5.34	1.95 ± 0.76
Social withdrawal	12.31 ± 4.72	2.46 ± 0.94
Public stigma	15.69 ± 6.37	1.74 ± 0.70
Family stigma	10.85 ± 4.54	1.80 ± 0.76
Infertility-related stress	150.03 ± 17.51	3.26 ± 0.38
Social concern	36.75 ± 5.62	3.68 ± 0.56
Relationship concern	29.42 ± 5.43	3.94 ± 0.54
Need for parenthood	31.91 ± 5.54	3.19 ± 0.56
Rejection of childless lifestyle	25.59 ± 4.84	3.20 ± 0.61
Sexual concern	26.37 ± 4.55	3.30 ± 0.57

**Table 3 healthcare-12-01053-t003:** Linear regression of stigma of women with infertility (N = 366).

Item	Unstandardized β	SE	Standardized β	*t*	*p*
Constant	14.129	25.420	——	0.556	0.579
Location of residence	−7.383	3.521	−0.114	−2.097	0.037 *
Regard children as the most important thing in life	−12.509	3.379	−0.186	−3.702	<0.001 ***
Talk to others about infertility	7.680	3.574	0.104	2.149	0.032 *
Infertility-related stress	0.553	0.091	0.295	6.101	<0.001 ***

Note: R^2^ = 0.191, after adjusted R^2^ = 0.173, *F* = 10.539, *p* < 0.001; * *p* < 0.05, *** *p* < 0.001.

## Data Availability

The datasets used during the current study are available from the corresponding author on reasonable request.

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
