# Peer review of "The Stigma and Infertility-Related Stress of Chinese Infertile Women: A Cross-Sectional Study"

_healthcare, 2024, doi:10.3390/healthcare12111053_

Round 1

Reviewer 1 Report

Comments and Suggestions for Authors

I would like to congratulate the authors on a very interesting paper.

The panel on Chinese infertility differs from the picture in Latin America when the patients of 2 tertiary hospitals show a very different age comparing to Latin America, where 74% of women are over 35 years of age, with similar schooling, considering we have  mostly private centers, and our patients are primarily urban. However, the stigma seems to me to be exactly the same, with the same social difficulties. This is an excellent point of how much infertility is  a burden!

No doubt about the importance of talking about infertility, But it seems to me that there is  a previous fundamental role in family counseling talk, in general.

Psychological interventions are always welcome, as well as the formation of groups in waiting rooms, a welcoming environment and good listening by the teams. And, let's consider that men should be included in this information and actions, since they account for 50% of the etiologies.

 Regards

Reviewer 2 Report

Comments and Suggestions for Authors

Dear colleagues,

Thank you very much for submitting your manuscript. 

The study concentrated on the stigmatisation of infertile women. Here is evidence that medical experts should help infertile women select the right course of action to reduce their stress levels both mentally and physically. The subject matter is intriguing and pertinent to the journal's field. It's simple to read, and the wording is precise. The manuscript's description and organisation are superb. Overall, the work is well-written and well-organised. The reviews of relevant literature are perceptive and educational. The tables are presented neatly and are simple to read and comprehend. The presented aspects sufficiently support the conclusions. The literature reviews are insightful and informative.

I congratulate all the authors for their efforts.

I have some minor concerns, which are described below:

- The first part of the Section Method is written: This study was approved by the ethics committee of the first author's university and tertiary-level hospitals, and participants were provided with informed consent. I find no evidence of ethical approvals.

- At the end of section 3.1. is written: Level of stigma and infertility-related stress in women with infertility. I think that is missing something in this affirmation.

- The abbreviation RMB is better to have an explanation for a better understanding for the reader.

Reviewer 3 Report

Comments and Suggestions for Authors

The aim of the current study is to determine the stigma and infertility related stress in Chinese infertile women using a cross-sectional study.

In methods:

The authors should explain about the explanatory and dependent variables and type of adjustment in the regression models. Which variables with what level of significance are put in the multiple regression model?

In results:

The Standard deviation of age should be written.

Data in table two should be revised. It seems the second column is the average score of sub-scales, not items. Also, in the first row, second column, the average score of each item for stigma means what? Stigma just has a total score that seems to be the total score of the ISS questionnaire. And also sixth row, second column for infertility-related stress.  

In Table 3, there are two B columns in the Table. It seems, one of them is Standardized B. The interpretation of “location of residence” is not clear. This variable is categorical, and without knowledge about the dummy code, the interpretation is not possible. For example, which level has the highest stigma? Also for other categorical variables in Table.

Also, for “Regard children as the most important thing in life” the beta coefficient is negative. It means, if the answer to this question is “yes” the score of stigma is lower than the “no” answer. In the discussion part, interpretation about this finding is reversed. Anyway, the interpretation depends on the dummy code of the variable. It should be mentioned.

Reviewer 4 Report

Comments and Suggestions for Authors

Dear Authors 

I would prefer to refer to the social origins and perceptions in China if these perceptions are universally applicable in the country.

The failure rate of these women regarding stigma is also important. Why was it not considered?

the relationship with the partner can also affect the mental state of the woman

Add these to the Limitations section and add to this section the research positives by converting the section as "limitation and strength" 

Good luck

Comments on the Quality of English Language

Minor editing
